# What Mechanisms Do Financial Marketization and China’s Fiscal Decentralization Have on Regional Energy Intensity? Evidence Based on Spatial Spillover and Panel Threshold Effects Perspectives

**DOI:** 10.3390/ijerph19095759

**Published:** 2022-05-09

**Authors:** Junbai Pan, Kun Lv, Shurong Yu, Dian Fu

**Affiliations:** 1Business School, Ningbo University, Ningbo 315211, China; pjb13579@163.com; 2Yangming College, Ningbo University, Ningbo 315211, China; yushurong1232022@163.com; 3College of Teacher Education, Ningbo University, Ningbo 315211, China; fd17815530502@163.com

**Keywords:** financial marketization, fiscal decentralization, regional energy intensity, systematic GMM model, spatial Durbin model, panel threshold model

## Abstract

Energy efficiency and energy intensity are gradually gaining attention, and it is now an important proposition to reconcile financial marketization, fiscal decentralization, and regional energy intensity. Using Chinese mainland provincial panel data (except Tibet) from 2007 to 2019, this study applied the dynamic panel system generalized method of moments model, the spatial Durbin model, and the panel threshold model to investigate the mechanisms of financial marketization and fiscal decentralization on regional energy intensity. The study found that financial marketization can play a significant role in suppressing regional energy intensity, while fiscal decentralization promotes energy intensity. Meanwhile, financial marketization in one province can have a negative spatial transmission effect on energy intensity in other provinces, while fiscal decentralization in one province has a negative spatial spillover effect on energy intensity in other provinces. Based on the analysis of the moderating and threshold effects, financial marketization not only moderates the negative externality of fiscal decentralization, making it inhibit energy intensity in the opposite direction, but also gradually increases the moderating effect on fiscal decentralization as the degree of financial marketization increases, showing a nonlinear inhibiting effect on regional energy intensity.

## 1. Introduction

In recent years, the world’s energy depletion, climate change, and environmental degradation have become increasingly troublesome. The question of how to achieve green growth has become a major concern for many countries [1]. The mission of lowering carbon emissions has also been a significant challenge to China [2], as a result of which China proposed a goal of “striving to peak carbon dioxide emissions by 2030 and attaining carbon neutrality by 2060” at the two sessions of the National People’s Congress in 2020 [3]. The underlying problem of the “carbon neutral” goal is the issue of energy, indicating that as China’s environmental carrying capacity decreases, energy security issues also emerge [4]. The traditional development model, which disregards energy losses, is no longer sustainable. On the one hand, it is necessary to save energy and avoid wasting energy in the production process. On the other hand, it is vital to improve the level of technology, promote regional innovation capacity, optimize and upgrade the industrial structure, and eliminate backward production capacity. In this process, scholars have paid close attention to regional energy intensity, reflecting a country’s or a region’s energy consumption per unit of output (E/Y) [5]. Energy intensity can also reasonably reflect the level of regional energy use and energy use efficiency. Many scholars also regard the suppression of regional energy intensity as a kind of Pareto improvement [6,7]. In other words, a decrease in energy intensity not only means an increase in the supply of public goods but also simultaneously indicates an increase in the level of regional productivity and the ability to achieve more output with less energy consumption.

Financial marketization has always been the direction of financial reform and development in emerging market countries such as China. With the continuous development of China’s economy and society, and the promotion of financial-market-oriented reforms in recent years, the financial market has taken shape and has had an impact on the regional environment and energy intensity. On the one hand, financial marketization has intensified competition among financial capital market players. At this stage, China’s financial system presents a situation where state-owned commercial banks actively or passively oligopolize the financial market. The oligopoly position of state-owned commercial banks comes from the “will of the governor” rather than market choice [8], reflecting the government’s excessive intervention in the financial market. Some government and government officials, tending to “compete for growth” for the sake of promotion, will invest more financial capital in traditional high-energy-consuming industries and “zombie enterprises” with simple technology and high short-term returns, and form a path dependence in order to pursue inefficient economic growth, which decreases the efficiency of regional energy use. The development of financial marketization, on the contrary, facilitates the dismantling of such oligopolies, promotes the rational allocation of financial resources, and curbs regional energy intensity. In the meantime, the process of clearing out “zombie enterprises” and eliminating traditional energy-intensive industries takes time, and the development of energy-efficient technologies in the short term requires higher costs and temporarily increases unemployment, so local governments, driven by short-term economic behavior, will not invest more resources in the research and transformation of energy-efficient technology industries. On the other hand, financial marketization also has the potential to exacerbate capital factor market distortions, thereby increasing the region’s energy intensity. The reason is that new industries are technologically complex, long-running, and risky, which is inconsistent with the investment logic of capitalists seeking short-term returns. As a result, it further entrenches the path dependence of traditional energy-intensive industries, thereby inhibiting the efficiency of regional energy use. Therefore, further research is needed on the impact of financial marketization on regional energy intensity.

In addition, many scholars have argued that institutional factors, represented by the issue of fiscal decentralization, can also profoundly affect regional energy use and environmental efficiency [9]. On the one hand, energy use cannot be improved without the government’s financial support. In the past, due to the tournament-style promotion system of officials and the performance appraisal system, local governments only focused on the economic benefits brought by the short-term high-energy-consumption model, which was not conducive to energy conservation and emission reduction, and to a certain extent, raised regional energy intensity. However, in recent years, China has gradually abandoned the system of using GDP growth as a performance assessment indicator for local governments. Some local governments have begun to use local finance to support the high-quality development of the regional economy, which provides institutional safeguards to curb energy intensity. On the other hand, fiscal decentralization is also closely related to changes in local industrial structure and production factor allocation, which in turn affect regional energy intensity.

Currently, China’s economic development is entering the “new normal”, and the central government is placing an increasing emphasis on improving the quality of economic development in its top-level design. Under the constraint of achieving the goal of “carbon neutrality”, the support of green economy industries is increasing while regional energy intensity is declining. However, in terms of local government decision-making behavior, the tournament-style system of “competition for growth” continues to play an important role. Emerging industries, due to their complex technology, long periodicity, and low short-term return, do not enjoy any preferential policy inclination or support compared to other traditional industries, which is contrary to the goal of high-quality economic development. As a result, it is vital to manage the interactions between financial marketization, fiscal decentralization, and regional energy intensity in order to maximize their advantages while sufficiently reducing energy intensity. Given the aforementioned issues, this paper used Chinese mainland provincial panel data (except Tibet) for the period 2007–2019 and introduced the spatial Durbin model and panel threshold model to study the relationship between green finance and energy intensity during the process of financial marketization in China’s provinces, as well as the spatial spillover effects of the three, in order to find policy ways to coordinate the three and promote the quality of economic development.

## 2. Theoretical Mechanism and Research Hypothesis

### 2.1. Financial Marketization and Energy Intensity: The “Important Tool”

In the Chinese context, financial marketization has three critical effects on regional energy intensity. Firstly, the “competitive liberalization impact” caused by the financial marketization expansion might attract more private capital, digital financial sectors, and financial technology goods to enter the financial market [10]. Thus, the oligopoly phenomenon, or the state-owned commercial banks oligomerizing the financial market that was initially formed for market and administrative reasons in China, could be broken [11]. The market competition will be intensified, and the original short-sighted investment behavior, where financial market players only care about short-term economic returns of investing in high-tech industries and tertiary industries to obtain long-term returns, will be transformed. The second effect of financial marketization on regional energy intensity is the “financial market regulation mechanism”, which is formed by the development of financial marketization; that is, a higher level of financial marketization means a lower degree of government intervention [12]. On the one hand, government intervention can lead to higher public R&D expenditures and increased social competition for innovation resources, leading to lower regional innovation capacity. On the other hand, excessive government intervention is detrimental to the elimination and clearing of backward production capacity due to “bottom-up competition” [13]. As a result, the growth and reform of financial marketization are helpful to avoid the negative externalities caused by government intervention and clean up energy-intensive businesses and “zombie companies” established by government intervention and path dependency, effectively reducing energy intensity. The final effect is that financial marketization can create an “aggregate investment effect” in favor of curbing energy intensity. The long payback period for R&D returns in emerging industries and technologies dictates that financial marketization has a high dependence on external capital input [14]. The development of financial marketization has enabled the rapid flow of capital and financial resources in the financial markets, accompanied by innovations in financial instruments and financial products [15], so that more and better financial resources can be rapidly concentrated in new industries that are more competitive at financing than “zombie enterprises”, which will promote energy efficiency and curb energy intensity. Based on the above three approaches, financial marketization is an “important tool” to curb energy intensity in China, and the following assumptions are made.

**Hypothesis** **1.***Financial marketization has the potential to considerably reduce regional energy intensity*.

### 2.2. Fiscal Decentralization and Energy Intensity: “Competition by the Bottom”

Chinese-style fiscal decentralization is non-demand-oriented and has an apparent political vertical centralization. In other words, the central government has absolute authority to appoint and appraise local officials [16], which frequently influences regional environmental efficiency and energy intensity by influencing local governments’ performance appraisals and official promotions [17]. At the same time, the current Chinese-style promotion tournament system directs officials to engage in short-term economic behavior and makes them less likely to invest limited financial resources in public goods. Thus, fiscal decentralization, while releasing endogenous incentives for local government officials to compete around economic growth [18], also supports and expands the size of energy-intensive industries within government jurisdictions [19]. The reason is that traditional energy-consuming industries are technologically mature, short-cycled, and can provide a short-term boost to regional economic growth for local officials to demonstrate their political performance. On the contrary, the high-tech and emerging industries are technologically complex, have long cycles, and are highly risky. Vigorously developing the industries is inconsistent with the logic of promoting the tournaments. Based on the study above, the following hypotheses are proposed:

**Hypothesis** **2.***Chinese-style fiscal decentralization can have a significant positive effect on regional energy intensity*.

### 2.3. Financial Marketization, Fiscal Decentralization, and Energy Intensity

The financial marketization process tends to have a moderating effect on fiscal decentralization and local government intervention. The reason is that the government’s fiscal spending and policy transmission need to rely on financial instruments and financial markets, among which macrocontrol and fiscal policies such as treasury bonds need to rely on financial markets for their transmission to market players [20]. Thus, with the development of financial marketization, the “financial market regulation mechanism” gradually replaces the local government’s allocation of capital factors. Due to the disintegration of path dependence formed by fiscal decentralization, the capital mismatch initially caused by China’s current fiscal decentralization system is also corrected, rapidly clearing out “zombie enterprises”, eliminating backward production capacity, and leveraging financial resources to cluster in emerging and high-tech industries. Additionally, fiscal decentralization is effectively regulated, making it difficult to suppress energy intensity, and this suppression will gradually increase as the level of financial marketization rises. After that, with the deepening of the financial marketization process, the “competition liberalization effect”, “financial market regulation mechanism”, and the “total investment effect” of financial marketization become the dominant components at different stages, respectively. Therefore, the adjustment effect of financial marketization on the negative externalities of fiscal decentralization shows a phased approach. That is, under the adjustment of financial marketization, the impact of fiscal decentralization on energy intensity shows nonlinear characteristics. Based on the analysis above, this study puts forward the following hypothesis:

**Hypothesis** **3.***Financial marketization has a moderating effect on the negative externalities of fiscal decentralization, and the two have a significant joint negative impact on regional energy intensity, which may be nonlinear depending on the economic and social development process in China*.

### 2.4. Spatial Spillover Effects: “Knowledge, Technology Spillover” and ”Crowding-Out Effects”

With the development of China’s market economy, interregional economic activities have become more frequent, and the flow of various economic factors between regions will inevitably result in the spillovers of economic effects, especially at a time when China’s financial market is becoming increasingly unified [21]. Financial resources can circulate freely between regions, so the positive externalities of financial marketization have spatial spillover effects. Furthermore, financial marketization can increase regional innovation through a variety of pathways and channels [22,23,24]. Regional innovation can reduce regional energy intensity and can also positively influence energy efficiency improvements in other regions, through knowledge and technology spillovers [25,26]. At the same time, an increase in the level of fiscal decentralization in a province can have a “crowding-out effect” on local innovative and emerging industries via a “bottom-up competition effect”, which has a positive externality effect on other provinces. As a result, under the Chinese fiscal decentralization model, increasing fiscal decentralization in one province will considerably reduce the energy intensity of other provinces. Similarly, the problems of irrational regional resource allocation, rising innovation costs, and diminishing regional innovation levels that accompany the increase in regional energy intensity would reduce the energy intensity of other areas by “crowding out” the province’s high-tech companies. This study suggests the following hypothesis based on the above analysis:

**Hypothesis** **4.***The development of financial marketization in a province can have a significant inhibiting effect on the energy intensity of other provinces, while the increase in energy intensity and fiscal decentralization level in a province has a significant negative transmission effect on the energy intensity of other provinces*.

## 3. Model Construction, Selection of Variables, and Spatial Distribution

### 3.1. Model Construction

#### 3.1.1. System GMM Model Construction

To examine the influence of financial marketization and fiscal decentralization on provincial energy intensity in 30 provinces of mainland China from 2007 to 2019, this paper added the first-order lag term of the explained variable regional energy intensity and built a dynamic panel GMM model.

At the same time, to solve the endogeneity problem in the model, this paper introduced the systematic GMM estimation method to obtain model 1 [27,28]. In addition, to study the relationship between financial marketization and the explanatory variables, fiscal decentralization, and further investigate the joint impact of financial marketization and fiscal decentralization on regional innovation, this paper multiplied the explanatory variables financial marketization (FM) and fiscal decentralization (FDE) to obtain a multiplicative interaction term as an additional explanatory variable in model 2.
(1)$$\mathrm{Model}\;1\text{: }EI_{it}=\Phi +\varphi EI_{i\left(t-1\right)}+\alpha_{1}FM_{it}+\alpha_{2}FDE_{it}+\sum \beta X_{it}+\varepsilon_{i\left(t-1\right)}$$
(2)$$\mathrm{Model}\;2\text{: }EI_{it}=\Phi +\varphi EI_{i\left(t-1\right)}+\alpha_{1}FM_{it}+\alpha_{2}FDE_{it}+\alpha_{3}GI_{it}\times GF_{it}+\sum \beta X_{it}+\varepsilon_{i\left(t-1\right)}$$

In the above equations, the regional energy intensity (EI) is the explained variable and financial marketization (FM) and fiscal decentralization (FDE) are the explanatory variables. $X_{it}$ represents the set of control variables. Φ is the constant term. φ is the regression coefficient of the first-order lag term of the explained variable regional innovation utility (EI).$\;\alpha_{1}$, $\;\alpha_{2}$, and$\;\alpha_{3}$ are the regression coefficients of the two explanatory variables and their interaction terms, respectively. β is the set of regression coefficients for the control variables.$\;\varepsilon_{t-1}$ is the residual term; i is the province, municipality, and autonomous region; **t** is the number of periods and **t − 1** is the lagged period.

#### 3.1.2. Spatial Econometric Model Construction

The spatial econometric model gives sufficient consideration to geographic unit dependency, which amends the defects of the classic econometric model. The development of financial marketization and fiscal decentralization of a region have an impact on neighboring regions or closely connected regions through spatial spillover effects, so this paper introduced a spatial econometric model to examine the impact of financial marketization and fiscal decentralization on regional energy intensity in each province of China from 2007 to 2019. In this study, we built a spatial autoregressive model (SAR), a spatial error model (SEM), and a spatial Durbin model (SDM) as shown in models 3 to 5, respectively [29].
(3)$$\mathrm{Model}\;3\text{: }EI_{it}=\rho W\cdot EI_{it}+\alpha_{1}FM_{it}+\alpha_{2}FDE_{it}+\sum \beta X_{it}+\gamma_{t}+u_{it}+\varepsilon_{it}$$
(4)$$\mathrm{Model}\;4\text{: }EI_{it}=\alpha_{1}FM_{it}+\alpha_{2}FDE_{it}+\sum \beta X_{it}+\gamma_{t}+\lambda W\cdot v_{it}+u_{it}+\varepsilon_{it}$$
(5)$$\mathrm{Model}\;5\text{: }EI_{it}=\rho W\cdot EI_{it}+\alpha_{1}FM_{it}+\alpha_{2}FDE_{it}+\sum \beta X_{it}+\theta W\left(EI_{it}+\sum \beta X_{it}\right)+\gamma_{t}+u_{it}+\varepsilon_{it}$$

In the above equations, the$\;GTPF_{it}$, $\;FM_{it}$, and$\;FDE_{it}$ represent the energy intensity, financial marketability index, and fiscal decentralization index of the **i**th province in year t, respectively.$\;X_{it}$ represents the set of control variables. ρ represents the lagged regression coefficients of the spatial explained variables, and θ represents the lagged regression coefficients of the spatial explanatory variables.$\;\alpha_{1}$ is the coefficient of the explanatory variable financial marketization index. $\alpha_{2}$ is the coefficient of the explanatory variable financial decentralization index.$\;\gamma_{t}$ represents time fixed effects.$\;u_{it}$ represents spatial fixed effects. W represents the spatial weight matrix.$\;\varepsilon_{it}$ represents the random error term.

#### 3.1.3. Spatial Econometric Model Construction

In light of the previous model’s endogeneity issues and in pursuit of a further exploration of the impact of green fiscal decentralization on carbon emission efficiency under the process of local financial marketization in China, this paper developed the following model 6 using the Hansen nonlinear static panel threshold model [30]. Model 6 is built using the financial marketization index (FM) as the threshold variable and fiscal decentralization (FDE) as the core explanatory variable.
(6)$$\mathrm{Model}\;6\text{: }EI_{it}=\Phi +\alpha_{1}FDE_{it}\times I\left(FM_{it}\leq \gamma \right)+\alpha_{2}FDE_{it}\times I(FM_{it}>\gamma )+\sum_{j}\beta_{j}x_{it}^{j}+\varepsilon_{it}$$

In the above equation, **i** denotes the province (**i** = 1, 2, 3 … 30), and **t** denotes time.$\;I\left(\cdot \right)$ denotes the indicator function, and γ represents the threshold value.$\;x_{it}^{j}$ denotes the set of control variables. $\alpha_{1}$and$\;\alpha_{2}$ represent the coefficients of the core explanatory variables at different stages.$\;\beta_{j}$ represents the coefficients of each control variable.$\;\varepsilon_{it}$ is the random error term.

### 3.2. Variable Interpretation and Data Sources

#### 3.2.1. Explained Variables

(1)Energy intensity (EI).

Energy intensity reflects the relationship between regional energy consumption and economic output and is usually used to represent the level of energy use and energy use efficiency. Referring to the study of Tajudeen et al. [31], this paper took the energy consumption ratio to the level of regional output (GDP) as the energy intensity level of the region. This indicator can be obtained by a direct query from the China Energy Statistics Yearbook in “tons of standard coal/yuan”.

The radar chart in Figure 1 is the spatial distribution of China’s provincial energy intensity in 2007, 2013, and 2019. It shows an overall decreasing trend during the reporting period and the uneven energy intensity levels among regions.

#### 3.2.2. Explanatory Variables

(1)Financial Marketability Index (FM).

In terms of measuring financial marketization, this paper referred to the “marketization index of the financial sector” presented in Fan et al.’s “China Marketization Index—Annual Report on the Relative Process of Marketization by Regions” as a proxy for financial marketization. The index was China’s only regularly released and province-specific financial marketization data and it included two aspects: market competition in the financial sector and marketization of credit fund allocation. Market competition in the financial sector is represented by the ratio of deposits taken by nonstate financial institutions to deposits taken by all financial institutions. Moreover, the marketization of credit fund allocation is represented by the share of deposits taken by all financial institutions. Among these, the degree of market competition in the financial sector fully reflects the degree of oligopoly and the level of entry barriers in the financial market. The marketization of credit fund allocation highlights financial capital allocation in the financial market. Above all, the index covers both capital competition and allocation dimensions in general.

Considering the fact that the current version of Fan’s China Marketization Index—Annual Report on the Relative Process of Marketization by Regions has adjusted the measurement base period compared to the 2011 version, as a result, the financial marketization index is not produced in the middle of the year. Referring to the method of Zhong et al., this paper applied the deflator method and interpolation method to the data. Finally, the paper gathered sample data for the paper’s primary explanatory variable, the financial marketization index. The radar chart in Figure 2 is the spatial distribution of China’s provincial average financial marketization index in 2007, 2013, and 2019. It shows that there was a general rise in financial market processes over the reporting period, but there was a lot of variation between regions.

(2)Fiscal Decentralization (FDE).

Fiscal decentralization refers to a certain degree of fiscal autonomy granted by the central government to local governments. Fiscal decentralization can be mainly divided into fiscal revenue decentralization, fiscal expenditure decentralization, and fiscal freedom decentralization. According to the study of Bai and Lu, the fiscal revenue decentralization indicator can be expressed as per capita on-budget fiscal revenue in the province/(per capita on-budget fiscal revenue in the province + per capita on-budget fiscal revenue in the central government); the fiscal expenditure decentralization indicator can be expressed as per capita on-budget fiscal expenditure in the province/(per capita on-budget fiscal expenditure in the province + per capita on-budget fiscal expenditure in the central government); and fiscal freedom can be expressed as budgetary revenue/budgetary expenditure.

Fiscal spending decentralization was used as a proxy measure of fiscal decentralization because it could not only reflect the government’s fiscal growth and the amount of government fiscal power but also be substantially connected with regional industrial development and energy usage. The China Statistical Yearbook and the statistical yearbooks of provinces and districts were used to acquire data on the central government’s, provinces’, and districts’ on-budget fiscal spending.

The radar chart in Figure 3 is the spatial distribution of China’s provincial average fiscal spending decentralization levels in 2007, 2013, and 2019.

#### 3.2.3. Control Variables

This paper selected forest cover (Forest), fiscal transparency (FT), industrial SO_2_ emissions (SDE), and capital mismatch (CM) as control variables for 30 provinces, municipalities directly under the central government, and autonomous regions (except Tibet) in mainland China from 2007 to 2019, based on the possible effects of financial capital allocation, fiscal policy, infrastructure, and industrial scale on regional energy intensity, fiscal decentralization, and the level of financial marketization.

(1)Forest cover (Forest): data from the China Statistical Yearbook and the statistical yearbooks of provinces and municipalities.(2)Fiscal transparency (FT): data on fiscal transparency are derived from the Fiscal Transparency Index reported in the Fiscal Transparency Report.(3)Industrial SO_2_ emissions (SDE): data from the China Statistical Yearbook and provincial and municipal statistical yearbooks, in million tons.(4)Capital mismatch (CM).

Referring to the factor relative mismatch measurement model proposed by Chen et al. [32] for measurement, capital mismatch was calculated using the model constructed as follows:(7)$$\mu_{it}=\frac{K_{it}/\sum_{i=1}^{n}K_{it}}{\left(Y_{it}/\sum_{i=1}^{n}Y_{it})(\alpha_{it}/\sum_{i=1}^{n}\alpha_{it}\right)}$$

In the above equation, $\mu_{it}$ represents the level of capital misallocation in period **t** in region **i**, and$\;Y_{it}$ represents the level of output in period **t** in region **i**.$\;K_{it}$ represents the capital allocated in period **t** in region **i**. $\alpha_{it}$ represents the output elasticity of capital in period **t** in region **i**.$\;\sum_{i=1}^{n}K_{it}$, $\sum_{i=1}^{n}Y_{it}$, and$\;\sum_{i=1}^{n}\alpha_{it}$ represent the country’s total capital allocation, total output, and capital-output elasticity, respectively.

According to the model, when$\;\mu_{it}>1$, it can be assumed that the cost of capital use is less than the national average when capital is underallocated; conversely, when$\;\mu_{it}<1$, it can be assumed that the capital is overallocated, which also means capital mismatch. Based on the relative mismatch of capital, this paper referred to the study of Ji [33] and introduced the absolute mismatch of capital to construct the model:(8)$$\lambda_{it}=\left|\mu_{it}-1\right|$$

$\lambda_{it\;}$ describes the absolute degree of distortion of capital, which was used in this paper as a proxy variable for the level of capital misallocation.

Table 1 reports the results of descriptive statistics for the explained, core explanatory, and control variables.

#### 3.2.4. Spatial Weighting Matrix

Since the research object of this paper may have spillover effects between regions with frequent economic exchanges or high economic linkages, this paper constructed an economic spatial weight matrix, which can better reflect the degree of economic linkages between regions. The matrix is shown in Equation (9).
(9)$$W_{i\theta }^{n}=\left\{\;\;\;\;\;\;\begin{array}{c} \frac{1}{\left|y_{i}-y_{\theta }\right|}\\ \;\;1\;\;\;\;\;\;\;\;\;i=\theta \;\; \end{array}\right.i\neq \theta$$

W is the spatial weight matrix and $W_{i\theta }$ is the element of the spatial weight matrix. $y_{i}$ and $y_{\theta }$ are the average real GDP per capita in **i** (1, 2, 3 … 30) regions and **θ** (1, 2, 3 … 30) regions, respectively, during the investigation period.

#### 3.2.5. Spatial and Temporal Distribution of Core Variables

Figure 4 depicts the spatiotemporal distribution of the core variables in 2007 and 2019 based on the preceding measurement and analysis of the key explanatory and explained variables.

Based on the above spatiotemporal distribution, the core variables of energy intensity, financial marketization, and fiscal decentralization show a significant spatial heterogeneity. The weight of financial marketization shows a “west-more, east-less” distribution pattern, while the energy intensity and fiscal decentralization show a “west-more, east-less” distribution pattern, which not only demonstrates the disparity in economic and social development levels, but the decentralization system between China’s east and west, but also highlights the need to consider the geographical heterogeneity of variable interactions in this study.

## 4. Empirical Results and Analysis

### 4.1. Dynamic Panel GMM Analysis

Model 1 and Model 2 were based on a systematic generalized method of moments (GMM) model, which was used to overcome endogeneity issues. Furthermore, the two models employed the autoregressive model test (AR) to assess the correlation of the disturbed term sequence and the Hansen test to determine the validity of the instrumental variables [34]. Table 2 and Table 3 report the results.

Model 1 passed the autoregressive model test (AR) and the Hansen test, as shown in Table 2. The coefficients of financial marketization in the conditions of different numbers of control variables were −0.0719, −0.0192, −0.0142, −0.0419, and −0.0165, which all passed the one percent significance test, indicating that the financial marketization process is positively correlated with the reduction in regional energy intensity and has a significant contribution to regional energy production efficiency. The reason is that, on the one hand, the financial marketization process is conducive to reducing the market oligopoly of state-owned commercial banks, accelerating competition in capital and financial markets [35], and compelling capital market participants to change their investment mindset. As a result, the players would redirect more financial resources previously invested in traditional energy-intensive industries for short-term gains to high-tech and clean production technology industries that are technologically complex, have high long-term economic benefits, and are supported by national policies. Financial marketization, on the other hand, implies less government intervention in the financial market, which can be beneficial in breaking the adverse effects of the government’s formalistic pursuit of economic growth figures under China’s tournament-style official promotion system, as well as quickly clearing out high-energy-consuming industries and “zombie enterprises” that exist in the market due to the collusion between government and enterprises and path dependence. Furthermore, financial marketization also increases capital liquidity in the financial market, which leads to an “aggregate investment impact” that allows more ways and channels for money to be invested in new industries, which improves energy efficiency.

At a 1% level of significance, the coefficients of fiscal decentralization were all significantly positive, with values of 0.508, 0.514, 0.619, 0.568, and 0.521, indicating that fiscal decentralization has a significant positive contribution to energy intensity and can reduce energy efficiency. This is primarily due to the irrational performance assessment and promotion systems. Under these systems, local governments and authorities are urged to focus on short-term economic growth [36], devoting more significant financial resources to conventional energy-intensive businesses rather than public amenities [37]. Thus, with more fiscal power, regional energy-intensive businesses might obtain greater financial and policy assistance, developing and deepening their route dependency, and ultimately falling into the trap of a “race to the bottom”.

In addition, in order to further investigate the joint mechanism of financial marketization and fiscal decentralization on regional energy intensity, the interaction term of the two was added as an explanatory variable in the model and the parameters were estimated again. The results obtained are shown in Table 3.

According to Table 3, the coefficients of the multiplicative interaction terms of financial marketization and fiscal decentralization are −0.259, −0.285, −0.296, −0.342, and −0.320, respectively. All pass the one percent significance level test, indicating that the joint main effect of financial marketization and fiscal decentralization on regional energy intensity is negative, demonstrating that financial marketization has a moderating effect on fiscal decentralization’s negative effects. The explanation behind this is as follows: to begin with, although fiscal decentralization has a significant negative impact due to the existence of a “bottom-up competition” with the development of financial marketization in recent years, local governments have gained more access to the market through financial instruments, investing in industries other than traditional energy-intensive industries; second, the growth of financial marketization has limited government interference, breaking the path dependency of old high-energy-consuming businesses, removing backward production capacity, increasing energy efficiency, and lowering energy intensity; third, the development of financial marketization has leveraged more scarce resources to gather in high-tech industries, promoting the transformation and upgrading of regional industrial structure, and innovating more “green finance”, “carbon-neutral”, and new industry-related financial instruments or financial products by intensifying financial market competition [38], which in turn significantly mitigates the negative externalities of fiscal decentralization and plays a positive role in public goods investment.

### 4.2. Spatial Econometric Model Analysis

#### 4.2.1. Space Suitability Test

To conduct a preliminary analysis of the model set’s rationality, this paper referred to the study of Long et al. [39] that used the global Moran index to verify the spatial clustering of regional energy intensity, financial marketization, and fiscal decentralization, as well as to test the variables’ spatial autocorrelation. Equation (10) is the measurement method of the global Moran index.
(10)$$Moran’s\;I=\frac{\sum_{i=1}^{N}\sum_{j=1}^{N}\omega_{ij}\left(x_{i}-\bar{x}\right)\left(x_{j}-\bar{x}\right)}{s^{2}\sum_{i=1}^{N}\sum_{j=1}^{N}\omega_{ij}},\;s^{2}=\frac{\sum_{i=1}^{N}{\left(x_{i}-\bar{x}\right)}^{2}}{N}$$

In the equation above, Moran’s I represents the global Moran index. N is the number of regions, and $x_{i}$ and $x_{j}$ represent the values of the variables in region **i** and region **j**, respectively. $\bar{x}$ represents the mean value of the variables and $\omega_{ij}$ represents the spatial weight matrix.

When the global Moran index is positive, the closer it is to 1, representing high–high clustering and low–low clustering, the stronger the spatial positive autocorrelation of the variable is considered; when the global Moran index is positive, the closer it is to 0, the more it is considered that there is no spatial autocorrelation; when the global Moran index is between −1 and 0, it is assumed that there is a negative autocorrelation of the variable.

Based on the economic spatial weight matrix, the global Moran indices of regional energy intensity (EI), financial marketability index (FM), and fiscal decentralization index (FDE) all passed the 1% significance test in the reporting period, indicating that all three were able to reject the original hypothesis, which means there was no spatial correlation at the 1% significance level for each year during the period 2007–2019. Therefore, there was a significant spatial correlation between the explained variables and the core explanatory variable. Figure 5 reports the overall Moran index of regional energy intensity (EI) and its *p*-value level for the period 2007–2019; Figure 6 shows the annual Moran scatter plots of regional energy intensity (EI) for the years 2007 to 2019. Both Figure 5 and Figure 6 justify the reasonability of the spatial econometric model set up in this paper under the economic spatial weight matrix.

#### 4.2.2. Identification, Selection, and Testing of Spatial Econometric Models

In the study, we chose the most appropriate spatial econometric model from models 3–5 to examine its plausibility. To begin with, this paper employed the approach of Le Sage et al. [29]: based on the OLS regression of the ordinary panel model, the Lagrange multiplier (LM) of the residuals and the robust form of this statistic (robust LM) were built to test for spatial autocorrelation. Table 2 displays the test results. Both the spatial autoregressive model and the spatial error model pass the LM statistic and its robust form under the economic spatial weight matrix. The spatial Durbin model (SDM) was used in this paper because it is the generic form of the spatial autoregressive model (SAR) and the spatial error model (SEM).

Then, this paper applied Hausman’s test based on the spatial Durbin model to validate the applicability of fixed and random effects. Next, the applicability of temporal fixed effects, spatial fixed effects, and double fixed effects on the spatial Durbin model was compared by constructing LR statistics. Finally, the Wald and LR tests were used to confirm if the spatial Durbin model was superior to the spatial autoregressive model and spatial error model. It was also confirmed whether the spatial Durbin model degenerated into the spatial autoregressive and spatial error models. Table 4 summarizes the above-mentioned identification and testing findings.

As demonstrated by the Hausman test, at a 1% significance level, the statistic contradicted the initial hypothesis of not employing fixed effects; hence, fixed effects were used in this paper. The LR test, on the other hand, rejected the initial hypothesis of temporal and spatial fixed effects at a 1% significant level; hence, double fixed effects were the best choices in this study. Furthermore, both the Wald and LR tests showed that the spatial Durbin model outperformed the spatial autoregressive and spatial error models and did not degenerate into the spatial autoregressive or spatial error models. In conclusion, the spatial Durbin model with fixed effects and double fixed effects was chosen as the method of analysis in this paper.

#### 4.2.3. Spatial Durbin Model Regression Analysis

Based on the identification, selection, and testing of spatial econometric models above, this paper analyzed the relationship between financial marketization, fiscal decentralization, and regional energy intensity using spatial Durbin models with fixed effects and double fixed effects. Table 5 reports the parameter estimation results of Model 5.

The spatial autoregressive coefficients in Table 5 are all significant at the 5% level of significance, and all values are negative. It is reasonable to assume that, at the moment, the geographical spillover effect created by a province’s increased energy intensity will result in a significant reduction in the energy intensity of provinces that are economically dependent on the province. The reason for this is that increasing a province’s energy intensity invariably results in a concentration of additional resources on backward production capacity represented by traditional energy-consuming industries, which has a “crowding-out effect” on new and high-tech industries, thereby promoting energy efficiency improvements in other provinces while decreasing their energy intensity.

Furthermore, the primary effect coefficients of the financial marketization index under varied numbers of control variables are −0.0913, −0.0945, −0.0918, −0.0827, and −0.0927, which all pass the one percent significance level test. Meanwhile, the fiscal decentralization index’s major impact coefficients are likewise significant at the 1% level, with values of 3.538, 3.469, 3.519, 3.340, and 3.714. This suggests that financial marketization may significantly reduce regional energy intensity, while fiscal decentralization can significantly increase regional energy intensity, validating the parameter estimation results of the dynamic panel system GMM model in the preceding section. It can be assumed that financial marketization effectively clears and eliminates backward production capacity in the market by breaking the oligopoly of state-owned commercial banks, reducing government intervention in the market, breaking the original market path dependence mechanism, and promoting rapid capital flow in the financial market, as well as leveraging scarce financial resources to high-tech and clean production technology industries, which improves energy efficiency. Fiscal decentralization, on the other hand, has resulted in a more significant concentration of financial resources in traditional energy-intensive industries in order to reap short-term economic advantages, diminishing energy efficiency and hastening the growth in energy intensity.

W × FM has coefficients of −1.026, −1.077, −1.085, −0.959, and −0.962. At the 1% significance level, they all reject the original hypothesis that they are equal to 0, indicating that financial marketization has a negative spatial spillover effect, i.e., the development of financial marketization in one province has a negative spatial transmission effect on the energy intensity of other provinces. This is due to the fact that the development of financial marketization not only leverages the concentration of financial resources in the province to high-tech industries, which promotes the upgrading of industrial structure and aids the province in reducing energy intensity, but also enhances the development of local, new industries, which in turn enhances technological innovation in other regions through knowledge spillover, thus creating a new capital accumulation structure in the province [40].

The coefficients of W × FDE are −24.57, −22.49, −22.51, −21.85, and −23.04, respectively. They all pass the 1% significance level test. Therefore, it can be assumed that a rise in one province’s fiscal decentralization has a considerable negative geographical transmission effect on the energy intensity of neighboring provinces. The reason is that fiscal decentralization, as a result of the “race to the bottom”, has prompted more resources in the province to concentrate on traditional high-energy and high-short-term economic benefits industries, resulting in a “crowding out effect” on high-tech and clean production technology industries, which already face difficulties in obtaining scarce development resources in the province. As a result, energy efficiency increased in other regions, so the spatial transmission effect of fiscal decentralization on energy intensity was significantly negative.

#### 4.2.4. Decomposition Analysis of Spillover Effects in the Spatial Durbin Model

To counteract the model estimation deviation in the point estimation tests of spatial spillover effects and accurately estimate spatial spillover effects, this paper decomposed the total effect into direct and indirect effects with the help of partial derivatives. Specifically, model 5 can be transformed as follows:(11)$$EI_{it}={\left(I-\rho W\right)}^{-1}[\alpha_{1}FM_{it}+\alpha_{2}FDE_{it}+\sum \beta X_{it}+\theta W(EI_{it}+\sum \beta X_{it})]+{\left(I-\rho W\right)}^{-1}\varepsilon_{it}$$

In the equation above, I is the unit matrix. Using the explanatory variable financial marketability index (FM) as an example, biasing Equation (11) to the explanatory variable financial marketability index (FM) yields:(12)$$[\frac{\partial \left(lnEI\right)}{\partial FM_{1t}}\cdots \frac{\partial \left(lnEI\right)}{\partial FM_{Kt}}]={\left(I-\rho W\right)}^{-1}\left[\begin{array}{ccc} \begin{array}{cc} \alpha_{11} & W_{12}\theta_{k}\\ W_{21}\theta_{k} & \alpha_{12} \end{array} & \cdots & \begin{array}{c} W_{1N}\theta_{k}\\ W_{2N}\theta_{k} \end{array}\\ \vdots & \ddots & \vdots \\ \begin{array}{cc} W_{N1}\theta_{k} & W_{N2}\theta_{k} \end{array} & \cdots & \alpha_{1K} \end{array}\right]$$

In Equation (12), the main diagonal coefficients of the matrix on the right-hand side of the equation are direct effects, representing the direct effect of the local explanatory variables on the local explanatory variables. The non-main-diagonal coefficients are indirect effects, representing the spatial spillover effects of the explanatory variables from other regions on the local explanatory variables.

In this study, Stata 15.0 was used to break down the spatial spillover effects of the spatial Durbin model for further analysis, and the direct impact, indirect effect, and total effect decomposition findings were produced. Due to space constraints, all control variables were included in the model for decomposition analysis in this study, and the results are shown in Table 6.

According to Table 6, the direct, indirect, and total effects of the financial marketization index on regional energy intensity are significantly negative. This result indicates that financial marketization not only has a direct positive effect on the energy efficiency improvement of a province, but also has a negative spatial transmission effect on the energy intensity of other provinces via economic activities. Thus, vigorously promoting the process of financial marketization can not only reduce the province’s energy intensity effectively, but also improve the energy efficiency in other provinces, forming a good regional synergy effect due to knowledge and technology spillover caused by the upgrading of industrial structure and the development of high-tech industries driven by the province’s financial marketization development.

The fiscal decentralization index’s direct effect coefficient is generally consistent with the main effect of the spatial Durbin model discussed above, i.e., fiscal decentralization promotes energy intensity in the province, demonstrating that the regional “race to the bottom” effect and the tournament-style promotion system of officials create local factor distortions and reduce energy use efficiency. Furthermore, the coefficient of the indirect effect of the fiscal decentralization index is significantly negative, indicating that an increase in fiscal decentralization in one province has a significant dampening effect on the energy intensity of other provinces, demonstrating the existence of the “crowding-out effect” once again.

#### 4.2.5. Robustness Test

(1)Exclude samples that may cause interference

The four municipalities of Beijing, Tianjin, Shanghai, and Chongqing were included in the study sample. Although these four cities are part of the provincial administrative divisions, they are major cities, or megacities, in terms of size. Furthermore, Shanghai, as China’s financial center, and Beijing, the “headquarters effect” of the financial market, are among the regions in China with exceptionally active financial markets. Additionally, the fiscal authority of these four cities differs from that of other provinces, which may interfere with the study of this paper. As a result, this paper removed the samples from Beijing, Tianjin, Shanghai, and Chongqing and re-estimated the parameters. The estimated conclusions were essentially compatible with the findings of this paper, proving that the study passes the robustness test. Table 7 shows the parameter estimate results after the four samples were removed.

(2)Exclusion of first year data

In the reporting period of this paper’s study, in which the global subprime mortgage crisis broke out in 2008, the Chinese government adjusted the strength and measures of financial market regulation after this financial crisis, and adjusted the governance strategy of fiscal policy, especially the macroprudential regulation of the financial system [29]. This huge change in the strategy may have a profound impact on the process of financial marketization. Therefore, data prior to 2008 create some confounding effects on the results of this paper. Therefore, this paper excluded the 2007 data and re-estimated the parameters, and the results are reported in Table 8. The estimated results are generally consistent with the findings above and demonstrate the robustness of this paper.

(3)Consideration of omitted variables

In addition to the explanatory and related control variables mentioned above that affect the regional innovation utility of the explained variables, variables such as population density (square kilometers per person) (PD), real foreign direct investment (billion yuan) (FDI), environmental protection expenditure (billion yuan) (EPE), and total retail sales of social consumer goods (billion yuan) (TRS) may all affect regional industrial structure changes by influencing them, changing the overall regional technology level, etc., directly or indirectly affecting regional energy intensity, which, as a result, interferes with the estimated results of the core explanatory variables. Therefore, these variables were added to the original model in turn and were re-estimated all together with the other variables one by one. Among the newly added variables, environmental protection expenditure (EPE) was obtained by referring to the “environmental protection expenditure” in the “Chinese Government Revenue and Expenditure Classification” and the statistical yearbook of each province and region. The results were generally consistent with the previous estimation results, which verified the robustness of the paper’s findings. In Table 9, only the estimation results after all omitted variables were added are reported.

#### 4.2.6. Extended Analysis

Given the historical reasons and different levels of social development of China’s major geographical subdivisions, the impact of government intervention and green finance on regional innovation levels may be characterized by regional atypicality. As a result, this paper divided the sample into two groups: eastern regions and midwestern regions. Then, the spatial Durbin model continued to be used to further investigate the impact of financial marketization and fiscal decentralization on the utility of regional innovation in order to explore the spatial inhibition of the subject of this paper. Due to length limitations, this paper only presents the parameter estimate results after all the control variables were added. The subregion parameter estimation results are reported in Table 10.

As shown by the parameter estimation results of the spatial Durbin model for the eastern region of China, the main effect parameter estimation results for the eastern region are basically consistent with the entire sample case. However, the spatial spillover effect of financial marketization is not significant in the parameter estimation results. The results indicated that the knowledge and technology spillover generated by financial marketization between regions disappears in the sample of the eastern region. This is due to the fact that the eastern region of China is the region with a higher level of economic and social development, and the level of technology and financial marketization are comparable between the regions. Thus, there are no significant spillovers of knowledge and technology between the eastern provinces and each other, but mainly on the level of social development and energy intensity in the midwestern regions.

As shown by the parameter estimation results of the spatial Durbin model for Eastern China, based on the economic spatial weight matrix, the spatial spillover effects of financial marketization and fiscal decentralization in the midwestern regions are not significant. The results indicate that under the current Chinese-style development model, the level of economic exchange and financial market integration in midwestern regions within the region is not high, and it is difficult for knowledge and technology spillover and industrial transfer to exist between regions. The spatial spillover effect of the midwestern regions is mainly manifested in taking over the knowledge spillover and industrial transfer from the eastern regions, receiving transfer payments from the central government, etc. Therefore, the spatial spillover effect within the region is not significant.

### 4.3. Panel Threshold Model Analysis

#### 4.3.1. Threshold Effects Test and Determination of Thresholds

To address the endogeneity issue and further investigate the mechanism between fiscal decentralization and financial marketization, as well as the influence of fiscal decentralization and financial marketization on regional energy intensity, this paper employed the fiscal decentralization index (FDE) as an explanatory variable and the financial marketization index (FM) as a threshold variable in a panel threshold model. Meanwhile, by using Stata 15.0 to self-sample 300 times, threshold degrees-of-freedom significance tests were conducted on the threshold variable financial marketability index (FM) under the assumption of no threshold effect. As indicated in Table 11, the single and double threshold effect tests in model 6 are significant, but the triple threshold effect fails the significance test, and the number of effective thresholds is two. Thus, the model should contain two thresholds.

#### 4.3.2. Threshold Authenticity Test

Figure 7 shows the results of the LR statistic, where Figure 7a shows the LR statistic for a model with a single threshold and Figure 7b shows the LR statistic for a model with a double threshold. The curves in the figure are the LR values. Additionally, the dashed line is the critical value of the LR statistic at a 5% significance level. The lowest value of the LR curve in the figure represents the LR value of the threshold financial marketability level (FM). The critical value is considered to be true when the LR value is lower than the critical value of the LR statistic. The results indicate that both the LR value for the single threshold and the LR value for the double threshold of FM are below the critical value. Therefore, the threshold is true.

#### 4.3.3. Analysis of Threshold Regression Results

After the threshold effect test, the fiscal decentralization index (FDE) passed the significance test under the threshold variable financial marketability index (FM), which proved the effect of fiscal decentralization on carbon emission efficiency had nonlinear characteristics. The results of each variable parameter estimation are shown in Table 12.

The following conclusions can be drawn from Table 12.

(1)Under the effect of the threshold variable financial marketization, fiscal decentralization has a significant inhibitory effect on regional energy intensity. Therefore, it can be assumed that the development of financial marketization significantly regulates the negative externality caused by the “bottom-up competition” effect of fiscal decentralization. Additionally, as the level of financial marketization increases, the absolute value of the coefficient of fiscal decentralization gradually increases as well. This shows that the dampening effect of fiscal decentralization on regional energy intensity increases with the process of financial marketization, which verifies the parameter estimation results of the previous systematic GMM model.(2)When the level of financial marketization is below the first threshold (4.4700), the moderating effect of financial marketization on fiscal decentralization is mainly based on the “competitive liberalization effect”. In other words, the development of financial marketization has broken the oligopoly of state-owned commercial banks in China’s financial market. As the competition in the financial market is intensifying, the change in investment thinking and investment patterns in the financial market has been promoted. In particular, private and Internet financing entry into the financial market has forced state-owned commercial enterprises to participate in market competition and carry out external pressure reforms [41]. The reforms at first were only some short-sighted investment behaviors that only cared about short-term economic returns. The reform then shifted to investing in high-tech and tertiary industries for long-term gains, putting it in a good position to compete in the long-term financial markets.(3)When the level of financial marketization is between the first threshold (4.4700) and the second threshold (5.8300), the regulation of financial marketization on the negative externality of fiscal decentralization is mainly based on the “financial market regulation mechanism”. In other words, the government’s financial resources usually enter the market through financial instruments and financial products. After the development of financial marketization to a certain level, the financial market uses its regulating mechanism to leverage financial resources to high-tech industries to support the improvement of energy efficiency. Moreover, the “zombie enterprises” formed as a result of path dependency are unable to compete with new industries for investment, financing, and capital to obtain more financial resources [42] and are quickly eliminated, further eliminating outdated capacity and curbing energy intensity.(4)When the level of financial marketization is above the second threshold (5.8300), the moderating effect of financial marketization on fiscal decentralization is mainly based on the “aggregate investment effect” [43]. In other words, when the development of the financial market reaches a certain level, the financial market shows higher activity. At the same time, the well-established financial system and financial market supervision system promote the diversification of financial instruments and financial innovation [44], providing more investment and financing channels for high-tech industries and tertiary industries to promote their development. Additionally, it enables high-tech businesses to obtain financial resources and capital support more quickly, as government interference in the flow of money and capital is reduced. This immediately increases energy efficiency and decreases the region’s energy intensity.

## 5. Conclusions and Policy Recommendations

### 5.1. Conclusions

The following is the paper’s conclusion based on the aforesaid analysis.

Through a dynamic panel system, GMM estimation, and spatial Durbin model main effects regression analysis, it was showed that financial marketization has a significant inhibitory effect on regional energy intensity, and financial marketization gives full play to the “grip effect” of improving energy efficiency and promoting energy conservation and emission reduction. At the same time, fiscal decentralization significantly increases regional energy intensity, indicating that under the “bottom-up competition” model, Chinese fiscal decentralization is not conducive to reducing energy intensity and improving energy efficiency.

By using the cross-products of the levels of financial marketization and fiscal decentralization as explanatory variables and conducting systematic GMM estimation, it was verified that financial marketization and fiscal decentralization have a significant negative impact on regional energy intensity, which shows that financial marketization can effectively moderate the negative externalities generated by fiscal decentralization and make it inhibit regional energy intensity inversely.

Through the analysis of the spatial spillover effects of the spatial Durbin model based on the economic spatial weight matrix and its decomposition results, it was proved that the development of financial marketization in a province significantly suppresses the energy intensity of provinces with which the province has closer economic ties through knowledge and technology spillover. At the same time, the increase in carbon emission intensity and fiscal decentralization level of a province has a negative external effect on the local area, while it has a significant negative spatial transmission effect on the energy intensity of other provinces through the “crowding-out effect” on the province’s high-tech and new industries.

Through the panel threshold effect analysis, it was once again proved that under the regulation of financial marketization, fiscal decentralization can significantly suppress regional energy intensity, while this suppression effect is nonlinear and gradually increases with the increase of the level of financial marketization. Additionally, this nonlinear effect is also formed in three stages with the effect of the “competition liberalization effect” of financial marketization, “financial market regulation mechanism”, and “aggregate investment effect”.

Through the analysis of the subregional spatial Durbin model, it was demonstrated that the spatial spillover effect of financial marketization in Eastern China would disappear. In other words, there is no significant knowledge and technology spillover within the eastern region of China. Additionally, the spatial spillover effect in the midwestern regions of China is not significant due to the lack of close intraregional economic exchanges.

### 5.2. Policy Suggestions

This paper provides the following policy suggestions based on its results.

To begin with, it is vital to maintain financial market liberalization while reducing government interference in financial markets. Financial marketization should be promoted indefinitely for the following reasons. To begin with, financial marketization can be an important tool for reducing energy intensity. Secondly, it has a large positive externality in optimizing resource allocation, provincial industrial structure, and promoting market competition. Thirdly, a mature financial market can better support sustainable economic development. To reduce the interference that governments have on financial markets, on the one hand, the reform process requires breaking the oligopoly status quo of state-owned commercial banks in the financial market, promoting orderly competition among capital players, forming an “aggregate investment effect”, and directing relatively more capital and financial resources to new industries in order to achieve low-carbon economic development. On the other hand, government intervention in the financial market should be reduced in order to break the path dependence formed by traditional high energy-consuming industries and “zombie enterprises”, as well as to eliminate the negative impact of government “bottom-up competition” on regional carbon emission efficiency.

In addition, the government should reform the performance appraisal system for officials, reducing the proportion of economic growth factors in the appraisal of officials, and incorporating more indicators such as energy use efficiency and regional innovation levels into the appraisal system. It is of great importance to build a scientific appraisal system based on the quality of economic development, avoiding “bottom-up competition” and guiding localities to engage in “top-up competition” as appropriate. In addition, given that fiscal decentralization can stimulate local officials to develop the regional economy, the central government needs to delegate authority appropriately. However, at the same time, the government also needs to give policy guidance to local governments while granting them a certain degree of fiscal authority to avoid bottom-up competition.

Finally, China’s eastern and midwestern regions must strengthen intraregional exchanges and cooperation, achieve regional knowledge and technology spillover and resource sharing, and promote integrated regional development. The government should also create a unified financial market across regions, fully exploit the regional spillover effects of financial marketization, improve regional energy efficiency, reduce energy intensity, and contribute to achieving “carbon neutrality” while maintaining national energy security.

## 6. Possible Research Contributions and Shortcomings

### 6.1. Possible Research Contributions

First of all, this paper incorporated financial marketization, fiscal decentralization and regional energy intensity into a unified research framework and integrated the dynamic panel system GMM model, the spatial Durbin model, and the panel threshold model to analyze the mechanisms of financial marketization and fiscal decentralization on regional energy intensity in the context of mainland China, providing new evidence for relevant studies.

Additionally, through the parameter estimation of the interaction term from the multiplication of the financial marketization index and the fiscal decentralization index, this paper further analyzed in depth the joint mechanism of financial marketization and fiscal decentralization on regional energy intensity in mainland China and provided new theoretical support for proposing relevant policies to coordinate the relationship between the three.

Furthermore, in the panel threshold effect analysis, this paper analyzed the nonlinear effect of fiscal decentralization on regional energy intensity under the role of financial marketization. In other words, with the advancement of financial marketization in mainland China, it was able to produce a “competitive liberalization effect”, a “financial market regulation mechanism” and a “total investment effect”, so that fiscal decentralization can play a positive role in the practice of reducing energy intensity.

In addition, through the spatial Durbin model analysis, this paper explored the spatial effects of financial marketization and fiscal decentralization on regional energy intensity based on the economic spatial weight matrix in the perspective of provincial panel data in mainland China, which on the one hand, avoided the neglect of the dependency between spatial units in previous related studies, and on the other hand, can provide policy recommendations for coordinating the synergistic development between different administrative regions in China to jointly improve regional energy use efficiency.

Finally, based on the development differences between eastern and midwestern provinces in mainland China, this paper conducted a subsample regression of panel data and provided customized policy recommendations for the development of different geographical subdivisions based on the spatial heterogeneity that existed in the findings.

### 6.2. Research Shortcomings and Future Research Directions

Firstly, this paper analyzed the effects of financial marketization and fiscal decentralization on regional energy intensity using panel data from 30 provinces in mainland China. An effective econometric analysis is inconceivable in China due to a lack of data disclosure and a small sample size at the city level. To address this issue, data collection was conducted via alternative channels in order to obtain support from additional microscopic evidence.

Secondly, this paper constructed a spatial econometric model based on the economic spatial weight matrix for analysis. Due to space constraints, the results of parameter estimation using the geographic distance spatial weight matrix, the neighboring geographic spatial weight matrix, and the human capital spatial weight matrix were not included in this paper. Future disclosure of these findings is necessary to ensure the study’s validity.

Thirdly, the Chinese government has stated explicitly that it intends to achieve “carbon peaking” and “carbon neutrality” by 2020. These new policies will undoubtedly have an effect on regional energy consumption levels. It is critical to continue studying the impact of China’s policies on the “carbon neutrality” target using sufficient panel data in the future.

## Figures and Tables

**Figure 1 ijerph-19-05759-f001:**
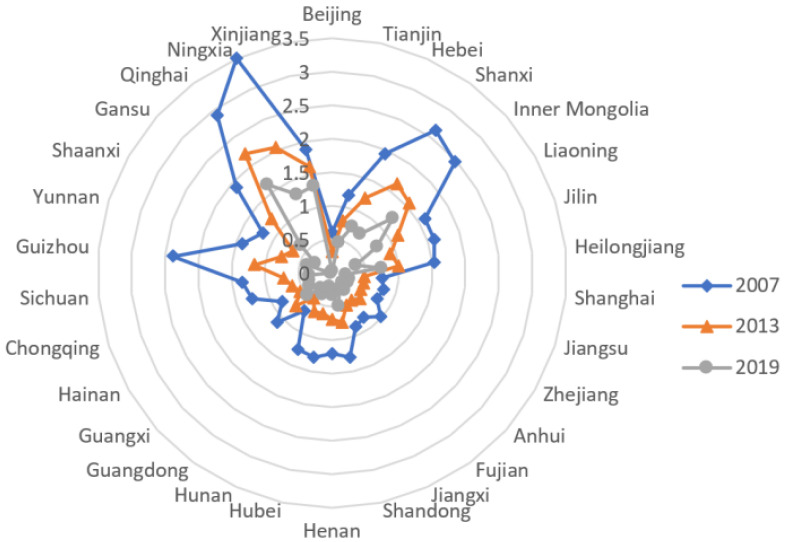
Radar chart of China’s provincial energy intensity.

**Figure 2 ijerph-19-05759-f002:**
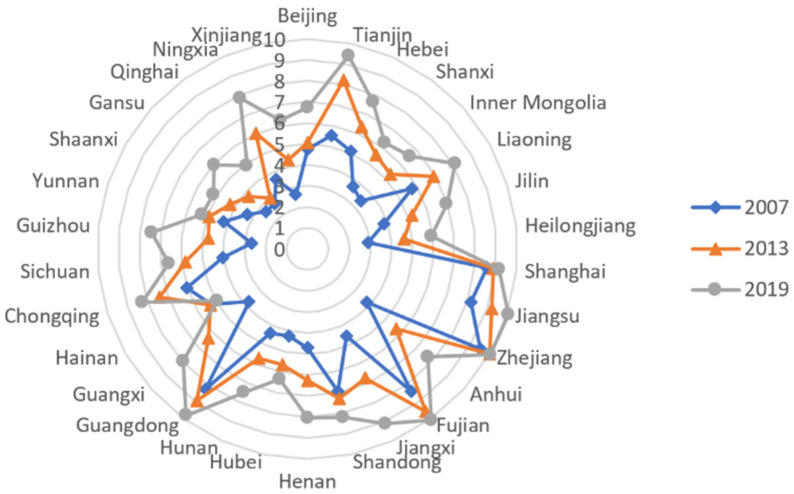
Radar chart of China’s provincial financial marketization index.

**Figure 3 ijerph-19-05759-f003:**
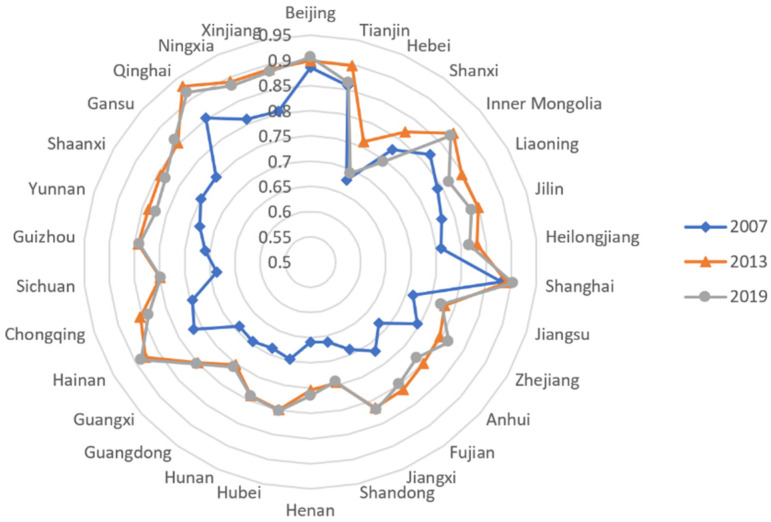
Radar chart of China’s provincial fiscal decentralization index.

**Figure 4 ijerph-19-05759-f004:**
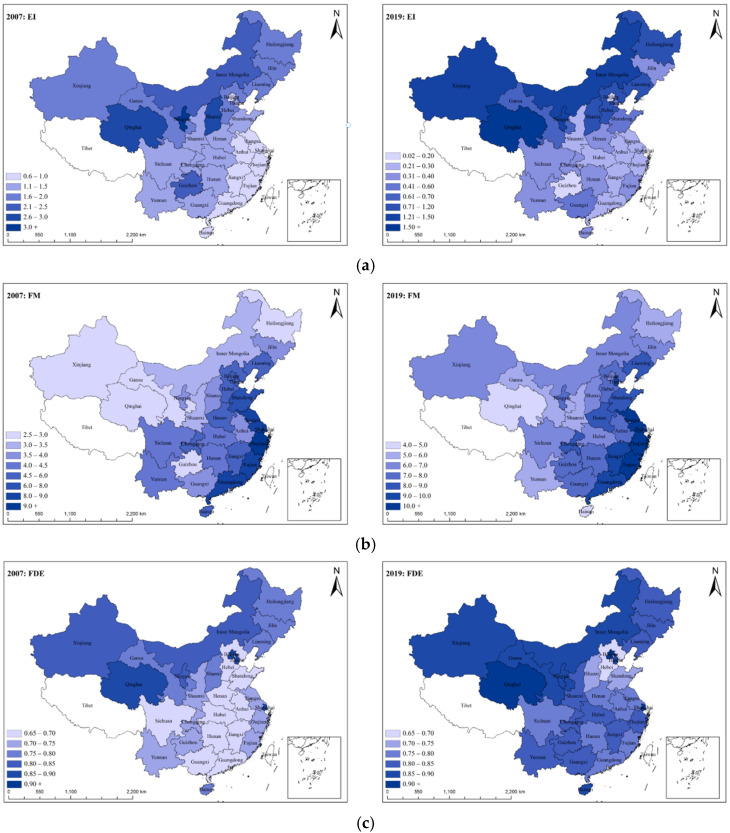
The spatiotemporal distribution of the core variables. (**a**) Spatial and temporal distribution of regional energy intensity. (**b**) Spatial and temporal distribution of financial marketization index. (**c**) Spatial and temporal distribution of the fiscal decentralization index.

**Figure 5 ijerph-19-05759-f005:**
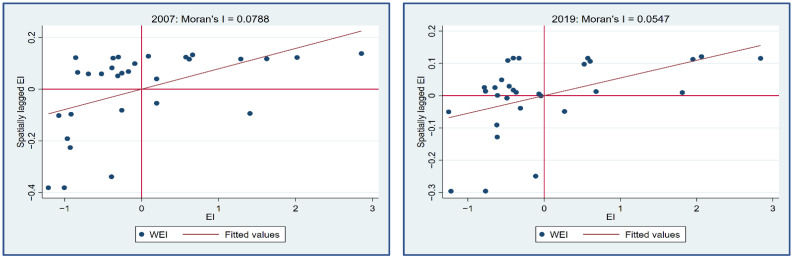
Overall Moran index and significance of energy intensity at the provincial level in China.

**Figure 6 ijerph-19-05759-f006:**
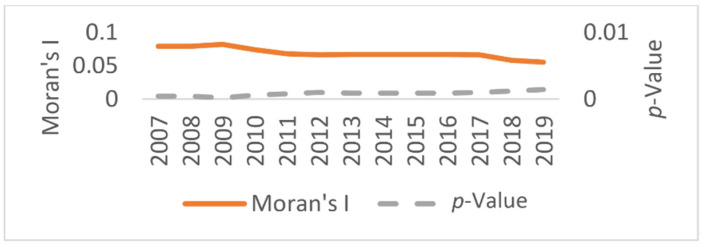
Annual Moran scatter plots of China’s provincial energy intensity from 2007 to 2019.

**Figure 7 ijerph-19-05759-f007:**
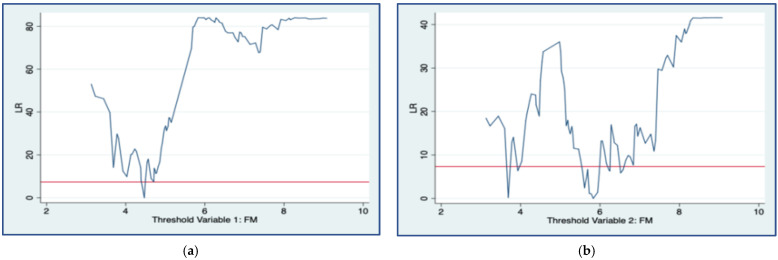
The LR value of the threshold financial marketability level (FM). (**a**) LR value of single threshold. (**b**) LR statistic of double threshold.

**Table 1 ijerph-19-05759-t001:** Descriptive statistics of the variables.

Variable	Connotation	Average Value	Variance (Statistics)	Minimum	Largest
EI	Energy intensity	0.964	0.573	0.021	3.506
FM	Financial marketization index	6.154	1.974	1.46	11.14
FDE	Decentralized index of fiscal expenditures	0.812	0.059	0.658	0.932
Forest	Forest cover	38.386	17.892	2.9	66
SDE	Industrial SO_2_ emissions (tonnes)	600,520.1	411,132.5	2800	1,800,000
FT	Financial transparency	33.793	18.495	1.12	109.7
CM	Capital mismatch	0.255	0.194	0.001	1.547
Sample Size	N = 390				
Period	2007–2019				

**Table 2 ijerph-19-05759-t002:** Parameter estimation results of Model 1.

	(1)	(2)	(3)	(4)	(5)
	EI	EI	EI	EI	EI
L.EI	0.884 ***	0.880 ***	0.854 ***	0.851 ***	0.847 ***
	(449.56)	(158.95)	(115.08)	(92.90)	(80.35)
FM	−0.0179 ***	−0.0192 ***	−0.0142 ***	−0.0149 ***	−0.0165 ***
	(−18.84)	(−9.71)	(−4.48)	(−5.32)	(−5.32)
FDE	0.508 ***	0.514 ***	0.619 ***	0.568 ***	0.521 ***
	(52.58)	(47.34)	(20.31)	(8.92)	(7.24)
Forest		−0.000125	−0.00103 ***	−0.000836 **	−0.00103 ***
		(−0.70)	(−3.22)	(−2.16)	(−2.76)
FT			−0.00168 ***	−0.00166 ***	−0.00203 ***
			(−17.55)	(−16.44)	(−10.72)
CM				0.000538	−0.000290
				(0.03)	(−0.02)
SDE					−3.43 × 10^−8^ ***
					(−3.56)
_cons	−0.265 ***	−0.254 ***	−0.259 ***	−0.213 ***	−0.123
	(−23.91)	(−12.14)	(−6.14)	(−3.51)	(−1.55)
N	360	360	360	360	360
*R* ^2^	—	—	—	—	—

t statistics in parentheses. ** *p* < 0.05, *** *p* < 0.01.

**Table 3 ijerph-19-05759-t003:** Parameter estimation results of Model 2.

	(1)	(2)	(3)	(4)	(5)
	EI	EI	EI	EI	EI
L.EI	0.904 ***	0.892 ***	0.862 ***	0.860 ***	0.856 ***
	(324.17)	(87.04)	(121.30)	(116.85)	(104.80)
FM	0.203 ***	0.222 ***	0.236 ***	0.274 ***	0.255 ***
	(12.51)	(11.89)	(8.55)	(6.16)	(5.47)
FDE	1.934 ***	2.009 ***	2.130 ***	2.340 ***	2.260 ***
	(25.31)	(23.70)	(12.01)	(9.15)	(8.44)
FDE × FM	−0.259 ***	−0.285 ***	−0.296 ***	−0.342 ***	−0.320 ***
	(−13.77)	(−15.61)	(−9.24)	(−6.49)	(−5.79)
Forest		−0.000638 ***	−0.00142 ***	−0.00169 ***	−0.00174 ***
		(−3.76)	(−3.81)	(−4.16)	(−4.40)
FT			−0.00178 ***	−0.00181 ***	−0.00176 ***
			(−20.44)	(−20.17)	(−15.84)
CM				0.00795	0.0310
				(0.34)	(0.99)
SDE					7.16 × 10^−9^
					(0.89)
_cons	−1.511 ***	−1.527 ***	−1.536 ***	−1.709 ***	−1.642 ***
	(−21.34)	(−14.84)	(−9.94)	(−7.82)	(−7.31)
N	360	360	360	360	360
*R* ^2^	—	—	—	—	—

t statistics in parentheses. *** *p* < 0.01.

**Table 4 ijerph-19-05759-t004:** Identification and testing of spatial econometric models.

Statistical Quantity	Numerical Value	*p*-Value
LM test no spatial lag	206.215 ***	0.000
Robust LM test no spatial lag	116.648 ***	0.006
LM test no spatial error	114.177 ***	0.000
Robust LM test no spatial error	24.646 ***	0.000
Hausman test	91.10 ***	0.0000
LR test for Time	630.49 ***	0.0000
LR test for Ind	68.31 ***	0.0000
Wald test for SAR	35.69 ***	0.0000
Wald test for SEM	27.45 **	0.0000
LR test for SAR	98.28 ***	0.0000
LR test for SEM	93.99 ***	0.0000

t-statistics in parentheses. ** *p* < 0.05, *** *p* < 0.01.

**Table 5 ijerph-19-05759-t005:** Parameter estimation results of spatial Durbin model.

	(1)	(2)	(3)	(4)	(5)
	EI	EI	EI	EI	EI
FM	−0.0913 ***	−0.0945 ***	−0.0918 ***	−0.0872 ***	−0.0927 ***
	(−4.78)	(−4.78)	(−4.58)	(−4.26)	(−4.52)
FDE	3.538 ***	3.469 ***	3.519 ***	3.340 ***	3.714 ***
	(5.48)	(5.33)	(5.36)	(5.09)	(5.47)
CM		0.00826	0.00370	0.0110	−0.00400
		(0.12)	(0.05)	(0.16)	(−0.06)
FT			0.000454	0.000626	0.000689
			(0.71)	(0.96)	(1.06)
SDE				0.000000113 *	7.00 × 10^−8^
				(1.82)	(1.07)
Forest					−0.0114 **
					(−2.01)
W × FM	−1.026 ***	−1.077 ***	−1.085 ***	−0.959 ***	−0.962 ***
	(−4.42)	(−4.43)	(−4.34)	(−3.61)	(−3.56)
W × FDE	−24.57 ***	−22.49 ***	−22.51 ***	−21.85 ***	−23.04 ***
	(−6.41)	(−5.73)	(−5.27)	(−5.11)	(−4.44)
W × CM		1.542 **	1.557 **	2.075 **	2.035 **
		(2.34)	(2.34)	(2.43)	(2.03)
W × FT			−0.00109	0.000620	0.000541
			(−0.12)	(0.06)	(0.05)
W × SDE				−0.00000142 *	−0.00000109
				(−1.67)	(−1.26)
W × Forest					0.0319
					(0.50)
Time fixed effect	yes	yes	yes	yes	yes
Area fixed effect	yes	yes	yes	yes	yes
Spatial					
Rho	−2.346 ***	−2.795 ***	−2.782 ***	−2.693 ***	−2.710 ***
	(−3.90)	(−4.09)	(−4.07)	(−3.99)	(−3.99)
Variance					
Sigma2_e	0.0161 ***	0.0163 ***	0.0163 ***	0.0162 ***	0.0160 ***
	(11.75)	(11.61)	(11.61)	(11.48)	(11.53)
N	390	390	390	390	390
*R* ^2^	0.507	0.517	0.516	0.512	0.668

t-statistics in parentheses. * *p* < 0.10, ** *p* < 0.05, *** *p* < 0.01.

**Table 6 ijerph-19-05759-t006:** Decomposition results of spillover effects for the spatial Durbin model.

	Direct Effect	Indirect Effect	Aggregate Effect
FM	−0.119 ***	−0.161 **	−0.280 ***
	(−8.17)	(−2.47)	(−4.51)
FDE	2.392 ***	−7.786 ***	−5.394 ***
	(5.51)	(−4.73)	(−3.47)
CM	0.0732	0.448	0.521 *
	(1.15)	(1.62)	(1.83)
FT	0.000662	−0.000354	0.000308
	(1.11)	(−0.13)	(0.11)
SDE	1.61 × 10^−8^	−0.000000268	−0.000000252
	(0.30)	(−1.02)	(−0.99)
Forest	−0.00886 **	0.0154	0.00655
	(−2.22)	(0.89)	(0.39)
N	390		
*R* ^2^	0.668		

t-statistics in parentheses. * *p* < 0.10, ** *p* < 0.05, *** *p* < 0.01.

**Table 7 ijerph-19-05759-t007:** Robustness test I: exclusion of samples that may cause interference.

Variable	Parameter Estimates	t-Statistic
FM	−0.104 ***	−4.53
FDE	4.177 ***	5.58
CM	0.323 **	2.27
FT	0.000631	0.89
SDE	6.06 × 10^−8^	0.88
Forest	−0.0100	−1.40
W × FM	−0.775 **	−2.20
W × FDE	−24.14 **	−2.52
W × CM	−2.871	−1.22
W × FT	−0.00736	−0.61
W × SDE	−0.000000538	−0.50
W × Forest	−0.0339	−0.32
Spatial		
Rho	−2.501 ***	−3.55
Variance		
Sigma2_e	0.0175 ***	10.71
N	338	
*R* ^2^	0.679	

t-statistics in parentheses. ** *p* < 0.05, *** *p* < 0.01.

**Table 8 ijerph-19-05759-t008:** Robustness test II: excluding the first year of data.

Variable	Parameter Estimates	t-Statistic
FM	−0.0915 ***	−4.56
FDE	3.540 ***	5.42
CM	−0.0101	−0.15
FT	0.000349	0.57
SDE	7.14 × 10^−8^	1.18
Forest	−0.0125 **	−2.16
W × FM	−0.960 ***	−3.83
W × FDE	−23.40 ***	−4.47
W × CM	1.630 *	1.77
W × FT	0.00515	0.56
W × SDE	−0.000000748	−0.95
W × Forest	0.0558	0.95
Spatial		
Rho	−2.384 ***	−3.64
Variance		
Sigma2_e	0.0122 ***	11.01
N	360	
*R* ^2^	0.664	

t-statistics in parentheses. * *p* < 0.10, ** *p* < 0.05, *** *p* < 0.01.

**Table 9 ijerph-19-05759-t009:** Robustness test III: considering omitted variables.

Variable	Parameter Estimates	t-Statistic
FM	−0.0743 ***	−3.70
FDE	2.458 ***	3.45
CM	0.0426	0.59
FT	0.000706	1.12
SDE	0.000000214 ***	3.09
Forest	−0.0106 *	−1.95
EPE	−0.000280	−1.34
FDI	−0.00000403	−0.09
PD	−0.0000417 **	−2.39
TRS	0.0000198 ***	4.02
W × FM	−0.816 **	−2.17
W × FDE	−14.14 **	−2.53
W × CM	0.688	0.50
W × FT	−0.00322	−0.30
W × SDE	−0.00000255 **	−2.14
W × Forest	0.0451	0.73
W × EPE	0.00254	0.84
W × FDI	0.00205 ***	3.02
W × PD	0.000212	0.59
W × TRS	−0.0000703	−1.11
Spatial		
Rho	−2.525 ***	−3.85
Variance		
Sigma2_e	0.0141 ***	11.70
N	390	
*R* ^2^	0.529	

t-statistics in parentheses. * *p* < 0.10, ** *p* < 0.05, *** *p* < 0.01.

**Table 10 ijerph-19-05759-t010:** Spatial Durbin model subregion parameter estimation results.

	Eastern Regions	Midwestern Regions
	EI	EI
FM	−0.0541 **	−0.178 ***
	(−2.24)	(−5.19)
FDE	3.645 ***	7.053 ***
	(5.76)	(6.56)
CM	−0.0353	0.209
	(−0.85)	(0.99)
FT	−0.000486	0.000755
	(−0.85)	(0.72)
SDE	2.55 × 10^−8^	0.000000258 ***
	(0.34)	(2.84)
Forest	0.0180 ***	−0.0136
	(3.13)	(−1.59)
W × FM	−0.207	0.157
	(−0.82)	(0.14)
W × FDE	−15.70 ***	−27.95
	(−4.87)	(−1.28)
W × CM	1.002	−7.429 **
	(1.55)	(−1.97)
W × FT	−0.00249	0.0291 **
	(−0.69)	(2.17)
W × SDE	−0.000000664	−0.000000671
	(−1.41)	(−0.32)
W × Forest	−0.0353	−0.316 **
	(−1.23)	(−2.34)
Time fixed effect	yes	yes
Area fixed effect	yes	yes
Spatial		
Rho	−2.330 ***	−2.608 ***
	(−3.11)	(−3.22)
Variance		
Sigma2_e	0.00350 ***	0.0178 ***
	(5.44)	(8.85)
N	156	234
*R* ^2^	0.009	0.663

t-statistics in parentheses. ** *p* < 0.05, *** *p* < 0.01.

**Table 11 ijerph-19-05759-t011:** Self-sampling tests for threshold effects.

Threshold Variables	Threshold Sequence	Threshold Value	*p*-Value	95% Confidence Interval	Number of BS	Seed Value
FM	Single threshold	4.4700 ***	0.0033	[4.3900 4.5000]	300	101
	Double threshold	5.8300 **	0.0100	[5.6450 5.9400]	300	101
	Three thresholds	7.3600	0.8533	[7.1450 7.4000]	300	101

t-statistics in parentheses. ** *p* < 0.05, *** *p* < 0.01.

**Table 12 ijerph-19-05759-t012:** Analysis of regression results for threshold effects.

Variable	Parameter Estimates	t-Statistic
Market	0.000000265 ***	3.14
FT	−0.00301 ***	−2.95
CM	−0.0943	−1.12
Forest	−0.0118	−1.35
FDE (FM < 4.4700)	−1.703 *	−2.03
FDE (4.4700 < FM < 5.8300)	−2.050 **	−2.55
FDE (FM > 5.8300)	−2.324 ***	−2.96
_cons	3.038 ***	5.79
N	390	
*R* ^2^	0.736	

t-statistics in parentheses. * *p* < 0.10, ** *p* < 0.05, *** *p* < 0.01.

## Data Availability

No new data were created or analyzed in this study. Data sharing is not applicable to this article.

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
