# Peer review of "What Mechanisms Do Financial Marketization and China’s Fiscal Decentralization Have on Regional Energy Intensity? Evidence Based on Spatial Spillover and Panel Threshold Effects Perspectives"

_ijerph, 2022, doi:10.3390/ijerph19095759_

Round 1
Reviewer 1 Report
In the introduction, it should be noted that the change to energy-saving technology entails high costs, a temporary increase in unemployment (especially if we intend to liquidate "zombie enterprises") and requires time.
The concept of monopoly and oligopoly cannot be used interchangeably - if state-owned banks compete with each other - it is an oligopoly.
The hypotheses should be more general - their wording indicates that the authors already know the results of the research.
On p. 12, one should rather use the expression that the process of financial marketization is positively related to the reduction of regional energy consumption (it is also influenced by a number of other factors).
Reviewer 2 Report
This paper presents and empirical study on the effect of financial marketization and fiscal decentralization mechanism on regional energy intensity by analysing Chinese mainland provincial panel data (2007-2029) The analysis has been carried out through Generalized Method of Moments (GMM) model, spatial Durbin model and panel threshold model. Further the spatial, knowledge and technology spillover effects of these mechanism in regional contexts have been shown.
In my opinion, the work is interesting for scientific community and provides a fair contribution in related research field.
The paper is well structured but there are numerous errors of sentence formation, repetition, and grammar, as so many times it’s not easily readable. Therefore, the manuscript must be thoroughly checked for language. Some part of the manuscript refers 'this report', should be changed with 'the present work' or 'this paper'.
Reviewer 3 Report
This study is interesting and sophisticated, but it needs extensive editing for proper English. For example, the very first sentence in the abstract is an incomplete sentence. I had to make many assumptions of what the intent was of many, many sentences.
R-squares are missing in Tables 2 and 3.
Reviewer 4 Report
The abstract should be more concise, organized in presenting the main aim of the paper, its objectives, the research method and the findings, but only briefly. Also, the abbreviations should be inserted first time in the text but not in the abstract. Avoid them in the abstract.
The introduction and literature review are well elaborated, also the methodology used.
In the conclusion section, I would avoid the numbering of ideas. Also, try to add some paragraphs about theoretical and practical implications of your research, limitations and future research directions and a concluding idea about the novelty of your study.
For the formatting part of the text, I would not leave blank spaces, you have almost half of page before some tables. Try to add text if it is not enough space for the table.
Otherwise, great success with your research.
